# Detection of Multi-Modal Doppler Spectra. Part 2: Evaluation of the Detection Algorithm and Exploring Characteristics of Multi-modal Spectra Using a Long-term Dataset

Sarah Wugofski<sup>1</sup> and Matthew R. Kumjian<sup>1</sup>

Correspondence: Sarah Wugofski (sjw5417@psu.edu)

Abstract. In this paper, we process three years of vertically pointing Ka-band radar spectral data according to the methodology described and established in Part 1 (Wugofski et al. 2025). Across three years of data, we demonstrate that the detection algorithm is successful in identifying multi-modal spectra, with 89.6% of detected events verifying. Beyond the verification, we explore other characteristics of the detected events such as the height, depth, and temperature of the layers containing secondary modes. Reanalysis data from ERA5 was used to gain additional context to the environmental conditions associated with the detected events. By connecting temperatures from ERA5 with the detected layers, we assess the potential for these events to be associated with common microphysical processes such as growth of columns or plates, Hallett-Mossop rime splintering, dendritic growth, and primary ice nucleation. We further explore the potential microphysical processes revealed by the multi-modal spectra using spectrally integrated linear depolarization ratio to determine if the secondary mode may comprise ice crystals that can produce such a signal. Of the cases with a detected enhanced LDR signal, >64.1% of those occurred in a layer with a mean temperature consistent with columnar ice or Hallett-Mossop rime splintering. Finally, three cases are investigated in more detail to illustrate the variety of events detected by the algorithm.

### 1 Introduction and Background

Remote sensing observations of cloud and precipitation, such as those from polarimetric radar, are useful in determining particle properties including size, aspect ratio, depolarization characteristics, and concentrations through variables such as reflectivity (Z), differential reflectivity (ZDR), linear depolarization ratio (LDR), and specific differential phase (KDP). In particular, exploring vertical changes in radar measurements – coined "fingerprints" (Kumjian et al., 2022) – provides information about changes to precipitation particles as they descend to the surface. When using vertically pointing radar, the mean Doppler velocity observation (MDV) can inform on particle fall speeds, vertical air motion, and/or the presence of turbulence and spectrum width (SW) can inform on the spread of MDV (see Part 1 and references therein). One of the most useful products from a vertically pointing radar is the Doppler spectrum,

<sup>&</sup>lt;sup>1</sup>Department of Meteorology & Atmospheric Science, The Pennsylvania State University, University Park, Pennsylvania, USA

which can be used for examining microphysical processes, including those when multiple types of hydrometeors are present. Doppler spectra, often visualized through spectrogram plots, show the distribution of returned power (or Z) across a range of Doppler velocities that can be often considered a proxy for particle fall speeds. Because different types of cloud and precipitation particles have varied sizes and masses, they have different fall speeds (e.g., Lamb and Verlinde, 2011), and thus their contributions to the Doppler spectrum often can be distinguished.

Spectral data contain particularly rich information for mixed-phase clouds, where particles such as cloud droplets, drizzle, ice crystals, and snow aggregates may coexist in the same radar sampling volume. Mixed-phase cloud processes are of particular interest because, for example, in the Arctic, mixed-phase clouds are long lived and cover large areas (e.g., Shupe et al., 2011; Morrison et al., 2012). As such, mixed-phase clouds have a large impact on radiative fluxes, which has implications for understanding climate impacts (Shupe and Intrieri, 2004; Zuidema et al., 2005; Morrison et al., 2012). Mixed-phase processes involve both liquid and ice hydrometeors, and can also include certain secondary ice-production mechanisms like Hallett-Mossop rime splintering (Hallett and Mossop, 1974) and droplet shattering (Field et al., 2017). These processes remain a great source of uncertainty in how we understand and represent the generation of ice particles, and so further investigations are needed (Field et al., 2017; Korolev et al., 2017). These processes can be investigated using radar Doppler spectra, particularly cases with multi-modal Doppler spectra (Luke et al., 2021; Billault-Roux et al., 2023).

In Wugofski et al. (2025) (hereafter Part 1), we show that multi-modal Doppler spectra from vertically pointing radars have a distinctive combination of large values of mean spectrum width  $\overline{SW}$  and small values of the standard deviation of mean Doppler velocity ( $\sigma(\text{MDV})$ ) over short (145 s long) data segments. In combination, these two quantities can be used to identify multi-modal layers, which were found to fall within a separate area of the  $\overline{SW}$  -  $\sigma(\text{MDV})$  parameter space compared to turbulent layers and non-turbulent, single-modal layers. An algorithm to detect the combination of these two quantities in vertically pointing radar moment data was created. Having established a proposed methodology for the detection of multi-modal spectra through radar moment processing, the algorithm can now be assessed for its ability to detect these events.

Here, we seek to evaluate the method proposed in Part 1 to identify multi-modal spectra events through analysis of radar moment variables. We test this algorithm using three years of data collected at the U.S. Department of Energy (DOE) Atmospheric Radiation Measurement (ARM) program North Slope of Alaska (NSA) site, and statistically evaluate its performance based on manual verification of the spectragraphs. We can then examine the temperatures of layers containing the detected multi-modal spectra and determine potential microphysical processes associated with the detected events. Although radar observations alone often are insufficient to conclude with certainty what processes are active in generating and growing observed hydrometeors, through observations of their fall speeds, depolarization signals, and proximal temperature profiles, we can assess how commonly the conditions favorable for such processes occur within this dataset.

#### 2 Data and Methods

75

85

#### 2.1 Radar and Algorithm

We apply the criteria established in Part 1 to three years of data collected by the NSA Ka-band ARM Zenith-pointing Radar (KAZR; see Part 1 for specifications). Specifically, we use the years 2020, 2022, and 2023. (Note that NSA KAZR data from 2013 were used in the development of the criteria, so we chose independent years on which to perform the evaluation.) Further, there were changes in data formatting in 2019; the CfRadial data format was adapted partway through 2019 and is currently used for KAZR data (Toto and Giangrande; Feng et al., a). For consistency, we use years after the change to the CfRadial convention. There is a significant gap in reliable KAZR spectra data in 2021 (March through October), in which data were negatively affected by artificial spectral broadening. The artificial broadening was likely caused by a malfunctioning phase lock oscillator that was ultimately replaced on 19 October 2021 (Min Deng, 2024, personal communication). Thus, 2021 is omitted from our analysis.

To process the long-term KAZR dataset, we partition the data into 145-s segments, matching those used for the algorithm development (Part 1). We keep the data segments a consistent duration because parameters (including the standard deviation) can change with increasing data temporal length. For each of these 145-s segments, we create vertical profiles of spectrum width (SW), mean Doppler velocity (MDV), signal-to-noise ratio (SNR), and linear depolarization ratio (LDR) with 30-m vertical resolution (matching the vertical resolution of KAZR). Points with a  $\overline{SW} > 0.17 \text{ m s}^{-1}$  and  $\sigma(\text{MDV}) < 0.1 \text{ m s}^{-1}$  are flagged for containing a potential secondary mode. Additionally, a SNR criterion is applied to filter out noise: data with SNR < -5 dB are omitted. Further, a filter to exclude data from in or below the melting layer is applied using the vertical gradient of LDR (see Part 1 for details).

Results of the detection algorithm (i.e., the "flagged" points) are then consolidated into detected cases. We set a minimum case duration of two hours to focus on persistent secondary mode signals. Although the present verification study considers cases of  $\geq 2$  hours to focus on more persistent signals, this choice can be made by algorithm users depending on the timescales of interest and tolerance for considering a greater number of events. Processes generating secondary modes can exist on short time scales, which was considered in the original determination of the use of 100 flags: the threshold for defining a case is set at 100 flags per hour, sustained for two hours. Several factors contributed to the choice of 100 flags hr<sup>-1</sup> as the threshold: with the 145-s and 30-m resolution of the NSA KAZR data, it takes approximately 100 flags to capture 15 minutes of a 0.5-km-deep multi-modal layer. This threshold of 100 flags hr<sup>-1</sup> can also be reached by an approximately 750-m deep multi-modal layer in just over 10 minutes. Thus, a case extending two hours should contain modes in both hours, though they may not continue for the entire duration of the event.

When examining the detected ("flagged") points in time-height space, we find they generally cluster into layers or streak-like features (Fig. 1). Whereas layers have a relatively constant height over time, streaks have a decreasing height over time. In some instances, there is speckling of flags that do not form a cohesive feature (e.g., as seen between 0-1 km and 4-5 km in Fig. 1b). In few cases, hourly flag counts were interrupted by an hour with >90

Figure 1. Examples of the detection algorithm output and associated radar moment variables for two events. (a) On 19 October 2020 the flags presented in streaks, beginning at higher altitude and dissipating at lower altitudes. (b) On 31 October 2024 the flags persisted in a layer. (i) Time-height depiction of radar gates that met the SW criterion (pink shading), the MDV criterion (blue shading), and where both criteria are met (black shading), indicating regions containing flags identified through the multi-modal detection algorithm. (ii) Radar reflectivity in dBZ. (iii) Spectrum width in m s<sup>-1</sup>. (iv) Mean Doppler velocity (i.e., scatterer vertical velocity) in m s<sup>-1</sup>.

but <100 flags; to avoid artificially inflating the case count, we consolidate these situations into a single case by considering them one event with start time of the first occurrence and end time of the last occurrence. Additionally, we observed sustained periods exceeding 100-200 flags hr<sup>-1</sup>, which will be further discussed in the results.

The detected cases are then manually verified through examination of instantaneous radar spectra, produced for every five minutes of a case. The secondary mode must be distinctly separated from the primary mode for the case to verify; we required at least a 5-dB decrease between the primary and secondary modes' peak values (see Part 1). During this step, the spectra are also checked for possible false detections due to turbulence or broadening from melting. In this way, we can compute the verification rate for the detection algorithm, as well as the false detection rate.

#### 2.2 ERA5

To better understand the forcings and processes associated with detected multi-modal events, ERA5 reanalysis (Hersbach et al., 2019) data complement the algorithm results and observed radar signals. Although there are upper-air observations from radiosondes taken at the NSA site, they are only routinely available twice daily, and thus often did not occur at the same time as the detected cases. For the detailed case analyses, we use the ERA5 hourly pressure-level data to extract thermodynamic information from a 1° × 1° box surrounding the NSA site (71.323°, -156.615°) for every other hour during the case. Data from within the box are averaged, and then we interpolate the vertical profiles from the 23 pressure levels to 90 height levels extending from 0 to 8.9 km in 100-m increments.

#### 2.3 Additional Instrumentation at the NSA Site

The NSA site has a wealth of additional instrumentation and numerous derived products that are useful for both considering the conditions associated with detected events. The microwave radiometer profiler (MWR) (Cadeddu et al., 2004) provides both precipitable water content and liquid water content. To construct a long-term climatology of the detected events and associated conditions, we use the MWR alongside derived products from the ARM Best Estimate Data Products (Chen and Xie, 1998) and Cloud type classification (Zhang et al., 1998). These products allow for comparisons to be drawn between the occurrence of a secondary mode and relevant conditions such as precipitable water vapor, liquid water content, cloud base and cloud top heights, cloud depth, and liquid equivalent precipitation. For considering individual case studies, we additionally use the precipitation imaging package (PIP) (Cromwell et al., 2018), which provides observations of the size and density distributions of hydrometeors observed at the surface.

## 3 Results

110

115

#### 3.1 Case Verification

The algorithm results can be examined in two frameworks: flag occurrence and case occurrence. We first discuss the cases. In considering the case counts, note that both 2022 and 2023 are influenced by periods of missing data. In 2022 there are short periods of missing data periodically from June through November of 2022 that may affect the total case counts in those months; these months contain 13-26 days rather than the expected 30-31. In July 2023, KAZR moment data were available but spectra were unavailable from 2-23 July. Across the three analysis years, 560 cases were found across 394 dates (Table 1). Of those cases, 89.6% (502) were verified by manual review of the Doppler spectra to have a secondary mode separated by local minimum in spectral power of >5 dB below the peaks. The verification rate was consistent across the three years, ranging from 86.0% to 92.5% (Table 1). 2020 had 186 cases, 172 of which verified. The fewest cases were identified in 2022, with only 158 verified out of 174 total cases.

Figure 2. Count of the number of verified cases by month, colored by year. 2020 is light green, 2022 is blue, and 2023 is purple.

2023 saw the most cases, with 200 detected events and 172 verified. Nonverifying cases have a separation of < 5 dB 0 between modes (i.e., they were less distinct but still multi-modal) or symmetric broadening of a single mode.

On average, 15.6 cases were detected per month, with 13.9 of those verifying as having secondary modes. When looking at all three years in aggregate, the months with the most identified cases are May, September, October, November, and December (Fig. 2). While some seasonal trends appear in this data, they are affected by the periods of missing data in 2022-2023 mentioned previously.

**Table 1.** Total number of dates flagged, total number of cases flagged and verified, and the algorithm success rate (correct detections, expressed as a fraction), by year and total across the 3-year period.

| Year  | Dates Flagged | Cases Flagged |          | Success Rate |
|-------|---------------|---------------|----------|--------------|
|       | Total         | Total         | Verified |              |
| 2020  | 130           | 186           | 172      | 0.925        |
| 2022  | 121           | 174           | 158      | 0.908        |
| 2023  | 143           | 200           | 172      | 0.860        |
| Total | 394           | 560           | 502      | 0.896        |

The median case length is 3 hr in duration (Fig. 3a); recall there is an imposed cut-off at two hours by the definition of a case set forth in the methodology (any single-hour cases have been excluded). The mean case duration is approximately 4.7 hr. These durations are suggestive of the timescales involved in the microphysical processes leading to the observed multi-modal spectra. The large frequency of occurrence of 2-4 hour multi-modal events, compared to sustained 5+ hour events, suggests that they are the result of processes that occur on shorter time scales. Despite most the of cases having durations < 6 hr, longer cases are present throughout the dataset. Examining cases lasting > 6 hr reveals they are more common in August and September (not shown, discussed more in 3.2).

These long-lived cases generally occur in the months with the most verified cases. Only 5% of verified cases last >12 hr, and the maximum duration observed was 23 hr. To investigate the case layer heights and depths, we use the 25th and 75th percentiles of flag heights to delineate the layer for each day. These are then investigated for dates that have verified cases. Detected multi-modal layers are present exclusively within the lower 5.3 km (Fig. 3b). The mean height at which these layers are detected is 1.9 km. The range of depths of detected multi-modal layers is quite variable, with the interquartile range spanning 0.69 to 1.56 km. The mean and median layer depths are 1.20 km and 1.14 km, respectively. Just over half of the verified multi-modal layers were shallower than 1 km, suggesting that processes creating and sustaining secondary modes are operating at a similar or shallower depth. Deeper layers are moderately correlated (Pearson correlation coefficient r = 0.52) with higher mean layer heights. Less than 12% of depths were >2 km; these deeper layers are likely explained by a combination of streak-like features that vary in height with time and dates with more than one distinct case present in a single day.

# 3.2 Flag Frequency Analysis

Examining the temporal distribution of flags across the three-year analysis period (Fig. 4 and Appendix) reveals visibly active times in which >100 flags hr<sup>-1</sup> are observed for extended durations. It is common for flagged hours to be clustered together as a multi-hour event, though some instances of isolated, single-hour periods with large flag counts do occur (e.g., 27 April 2020 at 14 UTC with 704 flags, and Figure 23 May 2020 at 19 UTC with 419 flags). In contrast, there are many multi-hour sustained events apparent, such as 30-31 July 2020. This case lasted 12 hr with an average flag count of 502 flags hr<sup>-1</sup>, minimum of 164 flags hr<sup>-1</sup>, and a maximum of 1039 flags hr<sup>-1</sup>. Similarly long cases with sustained flag counts exceeding 200 hr<sup>-1</sup> can be seen 15 April 2020 and throughout September 2020. While the cases with flag counts exceeding 200 hr<sup>-1</sup> stand out visually in Fig. 4, sustained cases with flag counts between 100-200 hr<sup>-1</sup> are also observable, such as 28 December 2020 from 13-15 UTC. The flag count for this event ranges from 115-136 hr<sup>-1</sup>. These results for 2022-2023 can be found in the Appendix.

To better understand the conditions associated with the detected events, we investigate how atmospheric liquid and cloudiness correlates with events using observations of precipitable water vapor from the MWR, liquid water path, cloud-base and cloud-top heights, and cloud thickness. We first analyze the relationship between flagged hours and hours that contain cloud cover. Note that this analysis examines all hours with a flag count of 100 or greater, not just events that see persistent signals of 2 or more hours. Because of extended periods of missing data in 2022-2023, only 2020 is used for this analysis. Cloudy periods are determined using the micropulse lidar (MPL) and radar-derived cloud fraction, with cloud fraction of  $\geq$ 75% being termed as a cloudy hour. Other fractional cloudiness thresholds were tested and the results were not significantly affected. This analysis found that, when using a cloudiness threshold of cloud fraction exceeding 75%, a total of 6321 of 8784 hours in the year were cloudy. 16.6% of those cloudy hours (1057 hours) were also flagged as exceeding 100 flags per hour (11.9% of all hours in the year). Interestingly, five hours were flagged that do not meet the cloudiness threshold.

Figure 3. Violin plots showing the distributions of (a) case duration (hr), (b) mean height of the detected layer (km), and (c) mean depth of the detected layer (km). The median and mean of the distributions are marked by black and blue stars, respectively. Vertical bars represent (from left to right): the minimum, the 5th percentile, the median, the 95th percentile, and the maximum.

Figure 4. Hourly flag occurrence for the year 2020. On each panel, the x axis is hour UTC and the y axis is day of month. Darker shades represent more flags occurring in a given hour, according to the color bar. The years 2022 and 2023 are shown in the Appendix.

First, we contrast observations of precipitable water vapor (PWV), liquid water path (LWP), cloud base, cloud top, cloud thickness, and hourly precipitation for hours with >100 flags hr<sup>-1</sup> and cloudy conditions and <100 hr<sup>-1</sup> and cloudy conditions (Fig. 5). The hours with greater than 100 flags hr<sup>-1</sup> generally had more moisture and deeper clouds than events with <100 flags hr<sup>-1</sup>. The cloud tops and thicknesses (Figs. 5d,e) were broadly spread for both the > 100 flag hr<sup>-1</sup> and < 100 flag hr<sup>-1</sup> cloudy periods. For cloud thickness, both interquartile ranges were approximately 4000-m wide, which is largely related to the variability in cloud tops. This large variability may be affected by multiple cloud layers, e.g., high-level cirrus clouds, rather than the tops of the specific cloud layers producing a multi-modal signal.

In Fig. 5c, there is a clear difference in the cloud bases of events with >100 flags hr<sup>-1</sup> and cloudy periods with <100 flags hr<sup>-1</sup>. The interquartile range (IQR) of >100 flag hr<sup>-1</sup> events cloud base is 84-1124 m, compared to 3189-4245 m for events with <100 flags hr<sup>-1</sup>. The average cloud-base height for >100 flag hr<sup>-1</sup> events is 942 m and <100 flags hr<sup>-1</sup> events is 1886 m. This potentially indicates that colder, higher clouds do not produce multi-modal signals and are not a significant portion of the detected events. This agrees well with the results shown in Figure 3, which, for events of 2 or more hours, showed a mean detected layer height near 1900 m and mean detected layer depth of approximately 1200 m. In Fig. 5f, there is a clear indication that hours with >100 flags have greater hourly precipitation rate. This is intuitive because the coexistence of multiple hydrometeor types or sizes (e.g., snow aggregates and cloud ice or graupel and liquid cloud droplets) would be expected in precipitation and expected to produce a multi-modal spectrum as the precipitation should typically has a faster downward vertical velocity (fall speed) than cloud liquid or cloud ice.

## 3.3 Comparison to Modality Detected by Gaussian Peak Fitting

185

190

205

195 To determine the success of this framework, we consider a comparison of detected multi-modal events to those detected by a Bayesian Gaussian Mixture model (GMM), as in Part 1. The GMM approach detects the number of peaks detected at a reduced temporal and spatial frequency to reduce the computational expenses needed to address a longer period of verification. We use the GMM approach to identify the number of peaks every 5 minutes with a spatial resolution of 90 m, using SciKit-learn (Pedregosa et al., 2011). We sample one week of data every three months during 2020, choosing the first week of that month containing at least one detected multi-modal event. This results in using the weeks of 8-14 January 2020, 1-7 March 2020, 1-7 July 2020, and 1-7 October 2020 for the verification analysis (Figure 6). This analysis allows for the identification of any missed detections in addition to false alarms as detected through the spectral verification described in Section 3.1.

From 5-16 UTC on 8 January, flags were present in low quantities, but did not meet the 100 flag hr<sup>-1</sup> threshold and both GMM and manual inspection of the spectra indicate that there was not an event at this time. Similarly, on 11-12 UTC on 13 January there was a two-hour period with few, but nonzero flag counts (70 at 11 UTC and 14 at 12 UTC) that were not detected by the GMM criteria. The remaining detected events coincide, although the GMM generally depicts the event as lasting longer than the duration of the flag count exceeding 100 flags hr<sup>-1</sup>.

Figure 5. Violin plots of observed quantities shown for hours in 2020 that contain 100 or more criteria flags per hour and hours in 2020 that do not meet this criteria, but do exceed 75% cloud cover. For events: the thick black line spans in the interquartile range and the thin grey line spans from the 10th to 90th percentiles. For no event: the thick green line spans in the interquartile range and the thin light green line spans from the 10th to 90th percentiles. (a) Precipitable water vapor as observed by MWR (b) liquid water path (c) cloud base (d) cloud top (e) cloud thickness (f) hourly liquid equivalent precipitation.

Figure 6. Four time series of week-long periods that were tested by both the described flag framework and through GMM peak detection. (a) 8-14 January 2020 (b) 1-7 March 2020 (c) 1-7 July 2020 (d) 1-7 October 2020.

There appears to be a false alarm detected by our algorithm on 0-4 UTC 1 April, wherein the flag count peaked at 518 for a single hour; however, this period had no other hours exceeding 100 flags and thus did not constitute an "event" in this work. Additionally, visual inspection of the spectra from 1-2 UTC reveals a bimodal layer present between 1-3 km AGL. Despite the occurrence of a bimodal layer, the GMM detection missed this event. There is a missed detection between 11 UTC 2 April and 2 UTC 3 April, in which there were multi-modal layers with shallow depths occurring periodically throughout this period. While there were criteria flags across this period, they fell short of the 100-flag hr<sup>-1</sup> threshold used to define a case (the greatest flag count was 87 hr<sup>-1</sup>). There is one more missed detection on 7 April, in which GMM identified a multi-modal layer that our algorithm missed. Upon inspection of the spectra for this case, the secondary mode had reflectivity near -10 dBZ for much of its life; this likely caused the mode to have a weaker effect on the integrated moment variables, leading to a missed detection.

Both of the weeks from July and October 2020 show good agreement between the criteria flag method and the GMM method. The 6 July 2020 event was depicted by the GMM as continuous across the first half of the day, whereas our algorithm indicates an interruption between 6-9 UTC during which the flag count falls to approximately 50 hr<sup>-1</sup>. This period of small flag counts is likely due to several factors, including the multi-modal layer being periodically

shallow, intermittently disappearing, and sometimes having a peak reflectivity of -15 to -10 dBZ when the primary mode was as strong as 10 dBZ.

This analysis comparing the two approaches reveals broad agreement between our algorithm and the more computationally expensive GMM approach. It is also apparent from this comparison that the occurrence of <100 flags hr<sup>-1</sup> can still be meaningful and suggests that an intermittent or weaker secondary mode is present during that period. Thus, investigators interested in exploring more subtle and transient multi-modal spectra may consider decreasing the hourly flag count criteria to better capture these events.

#### 230 4 Process Identification

235

Having demonstrated success in identifying multi-modal spectra through only the use of radar moment data, the next question arises: how do we determine the physical processes responsible for these features? Radar observations can provide insight into the shape, size, and concentration of hydrometeors, but ambiguity remains as to what processes are active within a cloud and responsible for the generation and growth of the observed hydrometeors.

Identifying the makeup of the primary and secondary modes is challenging. Observed modes may comprise ice or liquid. Recall that these modes are distinctly separated by their fall speeds and are commonly referred to based on their spectral power or reflectivity values: The primary mode is the one with the greatest reflectivity, and the secondary/tertiary modes have smaller reflectivity values. Primary modes often are the faster-falling mode, since greater fallspeeds usually implies more massive particles, and thus greater backscattering. Secondary modes often contain slow-falling ice generated from primary nucleation or secondary ice production processes, or small liquid cloud or drizzle droplets (e.g., Luke et al., 2010; Verlinde et al., 2013). To infer the microphysical processes responsible for the appearance of spectral modes, one important consideration is temperature: Certain processes are active in specific temperature ranges, such as dendritic growth from approximately -20 to -10 °C and Hallett-Mossop rime-splintering from -8 to -3 °C. The distinction between secondary modes containing liquid droplets or ice is commonly made using LDR (e.g., Oue et al., 2015; Sinclair et al., 2016). For vertically pointing radar, the LDR observed in columnar ice crystals is much larger than that of approximately spherical liquid droplets (e.g., Devisetty et al., 2019; Kumjian et al., 2020). One exception is melting ice hydrometeors, which can exhibit large LDR values (e.g., Li et al., 2020). Thus, to shed some light on the possible underlying physical processes responsible for the observed multi-modal spectra, we analyze both the temperatures and LDR associated with each verified case.

#### 250 4.1 Temperature

To analyze the temperature profiles associated with the verified cases, we use the 25th and 75th percentile flagged heights for each hour of data of each flagged case and use the ERA5 dataset to calculate the mean temperature and temperature range within the layer. These distributions are presented in Fig. 7. Note that the general climate of the NSA site analyzed here will affect the distribution of temperatures associated with detected multi-modal spectra

Figure 7. Violin plots showing the distributions of (a) mean temperature of the detected layer (°C), and (b) temperature range of the detected layer (°C). The median and mean of the distributions are marked by black and blue stars, respectively. Vertical bars represent (from left to right): the minimum, the 5th percentile, the median, the 95th percentile, and the maximum.

events and are likely unique to the Arctic climate of NSA and not applicable to multi-modal spectra events detected in mid-latitude or other environments. The mean temperatures center around -10.2 °C, with the interquartile range from -13.5 and -6.9 °C (Fig. 7a). However, the distribution features a long tail towards lower values, with some cases as cold as -28 °C. The temperature ranges associated with multi-modal layers are shown in Fig. 7b: >50% of the cases have a temperature range < 5 °C within the layer. Cases with temperature ranges >5 °C are attributable to deeper flagged layers. Manual inspection of the data revealed that some cases with fall streaks with vertical extents > 1 km are responsible for some of these larger temperature ranges.

We can partition the case mean temperatures into categories to access the favorability of certain processes such as rime splintering and dendritic growth. Though the full extent of the dendritic growth zone is often considered to extend from -20 to -10°C, we highlight the central temperatures of the dendritic growth zone from -18 to -12 °C, to maintain roughly equal-sized temperature categories as those of other processes. The dark blue bars in Fig. 8 show the percentage of cases that fall within the prescribed temperature ranges. To focus on mixed-phase and ice processes, we exclude 9% of the total cases that have an associated mean layer temperature  $\geq$ 0 °C. While the rain filter described in Part 1 and the Methods section is targeted at eliminating or reducing the detection of rain, melting and rain may be present in some of the warmer detected cases<sup>1</sup>.

#### 270 **4.2** LDR

265

LDR is useful in distinguishing liquid from columnar ice hydrometeors in vertically pointing radar data. The LDR moment data (i.e., integrated over the spectrum) associated with each flagged time and height is used for the analysis here. Because the moment LDR data comprise contributions from both the primary and secondary modes, LDR is not solely determined by the secondary mode. Thus, the moment LDR generally will not be as enhanced as the underlying secondary mode may be (particularly those attributable to pristine columnar ice). Due to maintenance of KAZR in 2021, the radar's lower LDR limit is significantly different in 2022-23 compared to 2020 (personal communication, Min Deng, 2024). To identify when cases have an LDR above the system limit associated with them, we first identify the average LDR associated with all cases within each year. We then add two standard deviations (calculated from this distribution of average LDR values from each year of cases) to this average to establish what threshold value we consider to be significantly greater than the average LDR. This results in the following LDR criteria: (i) for 2020: -17.96 dB, (ii) for 2022: -22.06 dB, (iii) for 2023: -21.62 dB. To determine what cases have layers with enhanced LDR values, we determine whether the 95th percentile flagged LDR for each case exceeds these thresholds. This was tested with both the 90th and 95th percentiles, which yielded a similar number of cases. This results in 26 days in 2020, 33 days in 2022, and 37 days in 2023 with multi-modal spectra events exceeding the LDR thresholds. For days with multiple distinct cases, we examine all cases on that date. These cases are then manually verified with spectral LDR computed from the co- and cross-polar radar spectra to determine which cases have secondary modes featuring enhanced spectral LDR values consistent with columnar ice crystals. Combining both LDR and temperature information, we find that 74.7% of the multi-modal spectra cases with enhanced LDR occur in a layer with mean temperature > -8 °C (Fig. 8; light blue bars), a disproportionately larger fraction than for all cases. There are two factors that may explain this result: melting is associated with increased LDR, and pristine columnar ice modes are associated with increased LDR. Only 10.6% of the cases feature mean temperatures >-3 °C, suggesting that melting is not the dominant contributor to these results. Instead, 64.1% of the multi-modal spectra cases with

<sup>&</sup>lt;sup>1</sup>Cases in the temperature distribution statistics presented in Figure 6 are verified multi-modal cases. Recall there is a criterion on MDV to exclude rain. Higher temperatures may, in part, be due to temporal or spatial displacement of the melting layer between the radar observations and modeled clouds in the reanalysis dataset.

Figure 8. Percentage of verified cases with mean layer temperatures  $<0^{\circ}$ C binned into five temperature categories defined on the x axis. Dark blue bars represent all verified cases with mean layer temperatures  $<0^{\circ}$  C and light blue bars represent only the subset of cases meeting the enhanced LDR thresholds defined in section 4.

enhanced LDR occur within the temperature zone favorable for Hallett-Mossop rime splintering (-3 to -8 °C) or primary nucleation of columnar crystals (e.g., Bailey and Hallett 2009). Only 7.7% of enhanced LDR cases occur in the -8 to -12 °C category, including the colder portion of the columnar habit temperature range and beginning of the dendritic growth range, suggesting that either primary generation of columnar ice or Hallett-Mossop rime splintering could be playing a role in the enhanced LDR subset of the detected cases. However, we cannot rule out other secondary ice mechanisms such as droplet shattering or collisional fragmentation, or new primary nucleation of columnar crystals (amongst extant snow and ice descending into this layer from above), due to lack of supporting in situ observations. In contrast, disproportionately few enhanced-LDR cases exist for temperatures lower than -12 °C, indicating that processes involving planar crystal habits or polycrystalline habits are not likely to produce secondary spectral modes with enhanced LDR values. More confident process attribution requires a more detailed analysis of individual cases, likely along with ancillary measurements. In the next section, we examine three cases in more detail to highlight the rich diversity of multi-modal spectral cases that can be identified using our proposed detection algorithm. We will examine cases within the warm (> -3 °C), columnar ice and Hallett-Mossop (-3 to -8 °C), and dendritic growth (-12 to -18 °C) temperature categories.

### 5 Selected Cases

# 5.1 Columnar Ice Growth and Rime Splintering Temperature Regime (-3 to -8 °C): 15 April 2020

On 15 April 2020, the algorithm detected a long-lived and deep multi-modal layer; the criteria were met from 1000 UTC 15 April through 0100 UTC 16 April. The detected mode had a mean height of 0.86 km and depth of 0.81 km. Over the 14-hour duration of this case, there was substantial variability in the reflectivity, downward velocities, and number of modes, including some periods featuring tri-modal spectra. For illustrative purposes, we narrow in on the hour of 2100-2200 UTC, and examine the integrated radar moments, the instantaneous spectra, and the flags associated with the 10-minute periods surrounding each time (Fig. 9 and 10). First, we consider the integrated radar moments to understand the context of the event. The mean detected layer height aligns well with the greatest reflectivity values observed. This layer has variable moment LDR values, ranging from -18 to -15 dB. There is a layer of spectrum width  $> 0.4 \text{ m s}^{-1}$  from 2130 to 2200 UTC above 1 km; the large values of spectrum width extend through lower heights from 2145-2200 UTC. The flags resulting from the detection algorithm generally align well with the multi-modal layers. Each 10-minute period contains 61 to 142 flags; even if only sustained for 20 minutes, such flag counts would meet the 100 flags necessary for progression through the analysis methodology. At 2200 UTC, the mean detected layer temperature was -4.13 °C, which falls within the temperature range for Hallett-Mossop ice splintering to be possible. Thus, we need to be aware of signals consistent with columnar ice crystals, such as increased LDR.

At 2105 UTC (Fig. 10a), the spectral reflectivity of both the primary and secondary modes are of a similar magnitude ( -10 dBZ; Fig. 11a). The slow-falling secondary mode increases in spectral reflectivity to +10 dBZ over the next 40 minutes (Figs. 10b-f, 11b-f), whereas the faster-falling primary mode's spectral reflectivity varies from -5 to 5 dBZ over the hour-long period (Figs. 10a-f; 11a-f). The size (or density) of the observed scatterers likely increases somewhat over the hour, as inferred from the secondary mode's downward velocity increasing from 0.25 to 0.5 m s<sup>-1</sup> in the first 10 minutes (Fig. 11a-b), after which it remains generally constant for the rest of the hour (Fig. 11c-f). These fall speeds are consistent with small hydrometeors such as ice crystals or small droplets. The large increase in spectral reflectivity coupled with the comparatively smaller changes in velocity of the slow-falling mode therefore suggests a rapid increase in the number concentration of the scatterers present, along with some growth in the particles' mass. The faster-falling mode's mean velocity increases from 1 to 1.5 m s<sup>-1</sup> over the hour (Fig. 11); these values suggest it could be snow aggregates or small graupel (e.g., Lamb and Verlinde, 2011; Jensen and Harrington, 2015; Heymsfield et al., 2018). Graupel would indicate that riming is present within this case; riming and graupel are required ingredients for rime splintering.

To understand the make-up of both the slower and faster falling modes, we consult the spectral LDR. At 2115 UTC, a majority of the slower-falling mode has LDR values near -14 dB, which is consistent with columnar ice crystals (e.g., Oue et al., 2015). At the same time, the faster-falling mode has LDR values near -20 dB, approaching KAZR's lower limit. These differences in LDR values between the two modes persist over the hour shown. As the

Figure 9. Spectrally integrated radar moments from an example case of multi-modal spectra on 15 April 2020: (a) reflectivity, (b) linear depolarization ratio, (c) spectrum width, (d) mean Doppler velocity. Areas with signal to noise ratio of less than 5 dB are masked out.

Figure 10. Example case of multi-modal spectra occurring in the temperature range that permits Hallett-Mossop rime splintering. Data are averaged from 10-minute periods centered on (a) 2105 UTC, (b) 2115 UTC, (c) 2125 UTC, (d) 2135 UTC, (e) 2145 UTC, (f) 2155 UTC. from 15 April 2020. Within each sub-panel, we show (left) waterfall plots of spectral co-polar reflectivity ZCO, (middle) waterfall plots of spectral LDR, and (iii) flags detected (blue bars) and a gaussian kernel density estimate of the distribution of flags with height (black line) are plotted. The black contours on the ZCO and LDR panels denote ZCO values > -10 dBZ, dark grey contours represent -20 dBZ. ZCO > -25 dBZ are masked out. The black horizontal lines in the ZCO panels show the height at which spectrograms are taken to analyze the modes (Fig. 11).

Figure 11. Spectrograms taken at 1 km AGL (denoted by the black lines in Figure 10) at (a) 2105 UTC, (b) 2115 UTC, (c) 2125 UTC, (d) 2135 UTC, (e) 2145 UTC, (f) 2155 UTC.

slower-falling mode's spectral reflectivity increases over time, larger LDR values are maintained at higher altitudes and on the slower-falling side (i.e., right edge) of this mode, whereas portions of the mode closer to the surface and closer to the faster falling mode exhibit lower LDR values closer to that of the faster-falling mode. This is likely a result of the ice crystals experiencing processes such as aggregation and riming as they descend. To support the potential for riming in this layer, we consider the liquid water content and precipitable water content from the MWR (Fig. 12). The liquid water content is greatest in the 0.5-1.0 km layer, with values as large as 0.4 g m-3. The liquid layer detected by the MWR is at the same height at the multi-modal layer: at least one peak through this hour is due to the presence of liquid droplets. The combination of liquid water droplets and LDR signal suggests that rime splintering is possible in this event.

# 350 5.2 Higher-Temperature Regime: 3 September 2020

345

A long-lived multi-modal event was detected on 3 September 2020 from 15 UTC – 3 UTC the following day. During this event, the lower-tropospheric temperature profile warmed over time: during the early hours of the event, the layer had a mean temperature of -5 °C, but by 0000 UTC 4 September, the layer-average temperature increased to +0.28 °C. To narrow our focus onto a case fitting the "warm" category established in section 4, we examine this multi-modal profile at 2345 UTC when the temperature of the layer was near 0° C (Fig. 13). The melting layer is located at 850 m; however, the spectra (Fig. 14) indicate the secondary mode extended both above and below the melting layer. Before 2340 UTC, the melting layer can be clearly seen in the enhanced LDR values. However after 2340 UTC there is a deeper region of increased LDR values that may be associated with columnar ice.

Across the hour shown in Fig. 14, the multi-modal layers are well co-located with the heights where flags were identified by detection algorithm. The heights of these layers vary throughout the hour, especially the low-reflectivity (-20 dBZ) modes at 2315-2325 UTC located above 1.3 km (Fig. 14b-c). By 2335 UTC, these become connected to the lower-altitude modes (Fig. 14d). For the purposes of this analysis, we will focus on the modes with spectral reflectivity values exceeding -10 dBZ.

The melting layer is just above 0.8 km, apparent from the increase in reflectivity and the rapid increase in magnitude and broadening of distribution of velocities (the dynamic range of raindrop velocities is nearly an order of magnitude greater than those of snow and ice; e.g., Lamb and Verlinde 2011). Note that the secondary mode persists below the melting layer; despite the faster-falling mode's downward motion exceeding 2 m s<sup>-1</sup>, the secondary mode is still detected by the flag criteria because the combined (reflectivity-weighted) MDV was still < 2 m s<sup>-1</sup>. This is most apparent in the snapshot at 2355 UTC (Fig. 14f) in which the secondary mode reflectivity exceeds 5 dBZ, allowing it to contribute enough to the moment variables to result in a MDV that does not exceed the rain filter criteria. This is contrasted with 2305 and 2325 UTC, where the secondary mode below the melting layer has a much weaker reflectivity and no secondary modes are flagged.

The evolution of the reflectivity and velocity of each mode is more easily quantified when examining spectrograms for this case taken at 1 km ARL (Fig. 15). At 2305 UTC, the spectrum is distinctly bi-modal, with an additional

Figure 12. Additional observations of precipitable water and liquid water content from the MWR and hydrometeor size distribution from the PIP on 15 April 2020 (a) precipitable water content (b) liquid water content (c) distribution of hydrometeor density vs. diameter.

Figure 13. Spectrally integrated radar moments from an example case of multi-modal spectra on 3 September 2020: (a) reflectivity, (b) linear depolarization ratio, (c) spectrum width, (d) mean Doppler velocity. This case represents the "warm" category.

Figure 14. As in Figure 9, for 3 September 2020 at (a) 2305 UTC, (b) 2315 UTC, (c) 2325 UTC, (d) 2335 UTC, (e) 2345 UTC, (f) 2355 UTC.

Figure 15. As in Figure 10, for 3 September 2020 at (a) 2305 UTC, (b) 2315 UTC, (c) 2325 UTC, (d) 2335 UTC, (e) 2345 UTC, (f) 2355 UTC.

75 indistinct peak on the slow-falling side of the primary mode. At all later times this hour, the spectra maintain three to four peaks. All times shown have the slowest-falling peak centered at  $0 \text{ m s}^{-1}$ . This is consistent with cloud droplets and/or particles suspended in an updraft.

From 2335 to 2345 UTC, the observed spectral reflectivity exhibits a quad-modal distribution (Fig. 14d, 15d). Beginning at 2335, the left edge of the fastest-falling mode broadens and shifts towards greater fall speeds. Although radar data alone are insufficient to determine the process(es) leading to the four distinct modes during this period, the end result is a significantly stronger primary mode at 2355 UTC with spectral reflectivity values 10 dBZ, much greater than 10 minutes prior (Fig. 14f, 15f).

Generally, the slower-falling modes in this case have relatively low spectral reflectivity (-10 to -5 dBZ); thus, because of the weak signal, LDR is positively biased and less reliable, making it more difficult to determine if the modes are caused by ice or liquid. Signals greater than -10-dBZ are more likely to be reliable than those signals associated with weaker returns. Thus, any LDR increase associated with Z decreasing to values below -10 dBZ are assumed to be biased and thus not used in the interpretation. However, at 2355 (Fig. 14f, 15f), when the secondary mode spectral reflectivity values exceed -10 dBZ over a 1-km depth, there is no enhancement of LDR and thus no associated depolarization signals to suggest that this mode contains ice. Rather, it is likely that this mode is composed of liquid droplets, likely drizzle drops due to the small fall speeds of 0 to 1 m s<sup>-1</sup>. The melting mode near 0.9 km has the greatest spectral reflectivity values observed in this case and is collocated with an enhancement of LDR. The slower-falling modes, for example at 2345 UTC, are weaker, with spectral reflectivity values < -10 dBZ. Because of the weak signal strength, we cannot infer the modes' composition using LDR. We can consider the precipitable water and liquid water content from the MWR to supplement this interpretation; there is liquid water present for much of this event, even above the melting layer (Fig. 16). This is consistent with our interpretation of the peaks centered at 0 m s<sup>-1</sup> in Figure 15.

## 5.3 Dendritic Growth Temperature Regime: 4 January 2022

The detected case on 4 January 2022 lasted four hours from 1300 UTC to 1700 UTC. This event had an average mean layer temperature of –13.7 °C, much colder than the two cases previously discussed and consistent with the dendritic growth layer (e.g., Lamb and Verlinde, 2011). First, we consider the spectrally integrated radar moments (Fig. 17). There are no notable signals in LDR (Fig. 17b), and the spectrum width is smaller than the previous two cases examined (Fig. 17c). Only a shallow multi-modal layer was detected by the criterion of a minimum of 5 dB difference from the primary mode (Fig. 18). The distinct layer is seen most clearly when viewing the 0-dBZ contour (Fig. 18) or the spectrograms taken at 2.5 and 3.0 km (Fig. 19). At 2.5 km, a secondary mode is still detected at all times shown, as indicated by the 10-minute flag counts (Fig. 18, right-most subpanels). However, the instantaneous spectra are more variable, with the secondary mode appearing less distinct at 1330 and 1340 UTC (Fig. 19b, d) and nonexistent at 1335 UTC (Fig. 19c). This illustrates the sensitivity of the detection algorithm to identify a secondary mode in a quickly changing radar presentation.

Figure 16. Additional observations of precipitable water and liquid water content from the MWR and hydrometeor size distribution from the PIP on 3 September 2020 (a) precipitable water content (b) liquid water content (c) distribution of hydrometeor density vs. diameter. Note the different y-axis scale in panel (c) compared to Fig. 12.

Figure 17. Radar moment variables from an example case of multi-modal spectra on 4 January 2022: (a) reflectivity, (b) linear depolarization ratio, (c) spectrum width, (d) mean Doppler velocity. This example is from the dendritic growth zone category.

Figure 18. As in Figure 10, with spectrograms taken at 2.5 and 3.0 km, for 4 January 2022 at (a) 1325 UTC, (b) 1330 UTC, (c) 1335 UTC, (d) 1340 UTC, (e) 1345 UTC, (f) 1350 UTC.

Figure 19. As in Figure 11, with spectrograms taken at 2.5 (blue) and 3.0 (orange) km, for 4 January 2022 at (a) 1325 UTC, (b) 1330 UTC, (c) 1335 UTC, (d) 1340 UTC, (e) 1345 UTC, (f) 1350 UTC.

The two modes in this case exhibit more similar velocities than those in the previous two cases: the primary mode varies from being centered on 0.8 to 1.0 m s<sup>-1</sup> while the secondary mode sits at about 0.5 m s<sup>-1</sup>. These fall speeds can be explained by a range of hydrometeor types, but small and/or less-dense snow aggregates are likely in the primary mode, especially given this layer having a temperature characteristic of the dendritic growth zone. The secondary mode may be explained by "early" aggregates (Moisseev et al., 2015) or pristine crystals (e.g., Lamb and Verlinde, 2011). Early aggregates are consistent with how the secondary mode extends towards the primary mode as it approaches the surface, similar to how an early aggregate may collect other crystals and increase its mass, and consequently, its fall speed. The MWR observations reveal little liquid water available at 2.5 km (Fig. 20), particularly compared to the previous two cases. Throughout this case, there is no clear enhancement in LDR for either mode that would be consistent with pristine columns or other ice crystals or crystal fragments with mass distributed asymmetrically in the horizontal plane. Thus, we speculate the multi-modal spectra arise from primary nucleation of planar crystals and their subsequent aggregation, in the presence of smaller aggregates falling into the layer from further aloft.

## 6 Conclusions

Following a three-year test of the multi-modal spectra detection algorithm described in Part 1, we found that it was 89.6% successful in identifying cases with secondary (and, at times, tertiary) modes. Using this moment-based detection algorithm will save users time and computational expense for processing large spectral datasets. Users looking for case studies of processes associated with multi-modal spectra, particularly those associated with mixed-phase or secondary ice production processes, can use this to identify dates and times of interest, narrowing down the number and size of radar spectra files needed. There may be merits to running this algorithm for similar long-term vertically pointing Ka-band radar datasets (e.g., the KAZR at DOE ARM sites) so that users can quickly identify these periods of interest. Applying a similar method on radars operating at other wavelengths may also be useful, though may require tuning the parameters used to compensate for differences in the radars themselves and differences arising from the different scattering responses of hydrometeors.

To further refine the criteria and explore potential processes associated with these modes, an LDR flag can facilitate finding cases specific to columnar ice or drizzle events. Although moment-based LDR criteria need to be accompanied by spectral LDR analysis to confirm the findings, the detection criteria can be helpful in narrowing down the pool of potential events. In the LDR analysis, we showed that nearly 60% cases meeting the criteria for having enhanced LDR were found in layers with temperatures commonly associated with both columnar ice crystal growth and Hallett-Mossop rime splintering (-8 to - 3 °C). Users interested in pursuing mixed-phase or secondary ice processes can use this as a springboard for further, in-depth analyses on LDR-flagged cases that can potentially confirm any processes active within the identified events. There are variable LDR limits for different radars, so further application of this approach to flagging enhanced LDR regions should keep in mind the potential variations in observed LDR by

Figure 20. Additional observations of precipitable water and liquid water content from the MWR and hydrometeor size distribution from the PIP on 4 January 2022 (a) precipitable water content (b) liquid water content (c) distribution of hydrometeor density vs. diameter.

different instruments. Our approach to identify events with an hourly mean LDR that was two standard deviations greater than the annual mean can be easily adapted to future studies with other radars.

Detected cases of multi-modal spectra within three temperature categories were examined: warm/near melting (> -3°C), Hallett-Mossop (-8 to -3°C), and the central range of the dendritic growth zone (-18 to -12°C). In all three cases, the algorithm detected criteria flags that aligned with the multi-modal layers. The warm case demonstrated that a bi-modal layer below the melting layer can still be detected using the algorithm, even with the rain filter. The three cases also illustrated that the detection is not limited to bi-modality, but that the algorithm will also identify layers that contain greater than two distinct modes.

While useful, radar alone is often insufficient to make concrete determinations of active processes. Application of this multi-modal spectra detection algorithm and LDR analysis benefits from accompanied analysis of atmospheric temperature profiles. Users can incorporate analyses of observational or reanalysis datasets to better understand what processes may be active in any detected multi-modal cases. Through LDR and temperature analysis, one may speculate about the active processes, but more in-depth process attribution is better supported by in-situ observations and particle imagery capable of confirming the presence, size, and concentrations of ice crystals, when available. Overall, this study has demonstrated the utility in identifying cases with processes capable of producing bi-, tri-, and quad-modal spectra via automation, which can be used to leverage large, archived radar data sets for new projects.

Code and data availability. KAZR moment and spectra data are available online at https://adc.arm.gov/discovery. ERA5 reanalysis data is available online through the ECMWF Climate Data Store at https://cds.climate.copernicus.eu/.

#### 460 Appendix A

As discussed in section 3, the temporal variation of flag counts is visualized in Fig. A1 and A2. Note that in 2022, the months of July through September include periods of missing data (5-18 days) containing no flags. The periods of missing data in Fig. A1 are apparent because, in 2020 and 2023, those periods contain higher flag counts and more long-duration events. For 2023 (Fig. A2), more flag activity is spread across all months compared to the previous years. Note that in 2023, July is excluded from verification due to missing spectral data from July 3-23; moment data are available, so flag counts are still depicted. While 2020 contains the same number of verified cases as 2023, the distribution of those cases across the months of the year is varied. The frequency and seasonality of these events over the three-year period, while providing a valuable perspective, are taken over a relatively short period. As such, for conclusive findings on the seasonality of multi-modal spectra events at the NSA site, a longer period of data should be considered.

Figure A1. As in Figure 4, but for 2022.

**Figure A2.** As in Figure 4, but for 2023. Results from July 2023 are displayed because vertically pointing moment data was available, but note that the spectra from this month was not available for verification.

Author contributions. This work stems from the dissertation work of SW, advised and guided by MRK.

Competing interests. The contact author has declared that none of the authors has any competing interests.

Acknowledgements. Support for SW and MRK comes from the U.S. Department of Energy Atmospheric System Research program Grant DE-SC0018933, the NASA Earth Venture Suborbital-3 (EVS-3) program under Grant 80NSSC19K0452, and academic appointments at The Pennsylvania State University. Data were obtained from the Atmospheric Radiation Measurement (ARM) Program sponsored by the U.S. Department of Energy, Office of Science, Office of Biological and Environmental Research, Climate and Environmental Sciences Division. We are grateful for the DOE ARM program's continued support of the KAZR facilities, data processing, and data storage that made this study possible. We would also like the thank Edward Luke for his helpful discussions about LDR and radar spectra.

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
