# Peer review of "Detection of Multi-Modal Doppler Spectra. Part 2: Evaluation of the Detection Algorithm and Exploring Characteristics of Multi-modal Spectra Using a Long-term Dataset"

_EGUsphere, 2025_

## Author Comment (AC1)

Reviewer #1

**General/Major comments:**

Major Comment #1: While there is an advantage of only relying on data from one instrument, KAZRs are usually not operated on a stand-alone basis but deployed with other instrumentation and the manuscript would greatly benefit from using these to substantiate the conclusions. – Currently, the potential microphysical processes leading to the observed multi-modal radar Doppler spectra like primary ice production, growth of column or plates, and Hallett-Mossop rime splintering are here only assessed based on ERA-5 temperature data instead of making use of the extensive ARM instrumentation suite (MWR, depolarization lidar, etc.). Adding observational evidence from other instrumentation would be very beneficial in interpretation of the temperature-only based hypothesis of microphysical processes. For example, to substantiate the hypothesis of potential occurrence of Hallett-Mossop rime-splintering, variables like liquid water path as derived from microwave radiometer observations help to see if liquid water was available for riming as prerequisite for rime-splintering. The points made in the manuscript would be much stronger if additional information from other instruments are added.

    Thanks for this comment, we agree and are incorporating this into the revised manuscript in a few different ways. First, we have added analyses of the precipitable water vapor and liquid water content from the MWR to the example cases. This is shown in figures R. 1, R. 2., and R. 3 below. The added context of the amounts of available liquid water and water vapor are useful for understanding the context of all three cases. In particular, we note that in the 15 April 2020 case within the temperature range favorable for Hallett-Mossop rime splintering (and columnar crystal growth), there is liquid water available to contribute to riming. For the warm case, 3 September 2020, the total precipitable water is greater than for any other case, though during a period of rain early in the hour there are missing observations. The final case, 4 January 2022, is the coldest and driest of those analyzed. These visualizations will be included in the sections pertaining to each of the cases.

[Figure]

Fig. R. 1: For 15 April 2020: (a) precipitable water vapor and (b) liquid water content for the hour of 21-22 UTC.

[Figure]

Fig. R. 2: As in Fig. R. 1, but for 3 September 2020 between 23-00 UTC.

[Figure]

Fig. R. 3: As in Fig. R. 1, but for 4 January 2022 between 13-14 UTC.

We additionally examined observations contained in the Precipitation Imaging Package, Microphysical Properties product (see nsaprecipipmpC1), particularly the observed hydrometeor sizes and densities at the surface. While this analysis is affected by observations being intermittent (15-30 minutes total of each hour), it allows readers to understand the precipitation created within these cases. This is shown in Figure R. 4. Both the averages and maximums are shown, as the averages are affected by the intermittent precipitation and data. 15 April 2020 has observed precipitation generally about 1 mm in diameter, with densities consistent with snow. 3 September 2020 has observed precipitation of a similar size, about 1 mm in diameter, but with densities consistent with liquid precipitation (rain). 4 January 2022 stands out as having larger observed hydrometeor size than the other cases, and with densities again consistent with snow. Given the knowledge that 4 January 2022 was a case within the dendritic growth zone, it follows that these observed particles may be larger because the dendrites grown aloft are forming snow aggregates.

[Figure]

Fig. R. 4: For cases (a) 14 April 2020 (b) 3 September 2020 (c) 4 January 2022: in black the average observed particle density and in green the maximum observed particle density.

Second, we have replaced the previous ERA5 climatology analysis with an analysis relating the flag counts with observations rather than a reanalysis. This addresses the same premise of testing to see if the flagged events are related to processes on a larger scale. This is discussed in depth in response to Major Comment #6.

With the additional datasets used to replace the original analysis as well as to supplement the cases examined in more detail, citations for additional ARM observational datasets and value added products (VAP) are attached:

Cadeddu, Maria, et al. "Microwave Radiometer Profiler (MWRP), 2004-02-19 to 2023-09-28, North Slope Alaska (NSA), Central Facility, Barrow AK (C1)." Atmospheric Radiation Measurement (ARM) User Facility, doi:10.5439/1025254. Accessed 25 May 2025.

Cromwell, Erol, et al. "Precipitation Imaging Package (PRECIPIPMP), 2018-10-23 to 2025-04-21, North Slope Alaska (NSA), Central Facility, Barrow AK (C1)." Atmospheric Radiation Measurement (ARM) User Facility, doi:10.5439/1489525. Accessed 25 May 2025.

Chen, Xiao, and Shaocheng Xie. "ARM Best Estimate Data Products (ARMBECLDRAD), 1998-01-01 to 2023-12-31, North Slope Alaska (NSA), Central Facility, Barrow AK (C1)." Atmospheric Radiation Measurement (ARM) User Facility, doi:10.5439/1333228.

Zhang, Damao, et al. "Cloud Type Classification (CLDTYPE), 1998-03-25 to 2023-08-30, North Slope Alaska (NSA), Central Facility, Barrow AK (C1)." Atmospheric Radiation Measurement (ARM) User Facility, doi:10.5439/1349884.

Major Comment #2: Line 289 - 290: In line 289 it is correctly stated that at T between -3 and -8°C both, primary ice nucleation of columnar ice crystals or Hallett-Mossop-rime splintering can occur. Please include the possibility of primary ice nucleation in this T-range in the abstract and in line 299 + line 314 etc. as well instead of only restricting to Hallett-Mossop-rime splintering. Furthermore, Section 5.1 should also be labeled accordingly.
This is very true! These corrections have been made and section 5.1 now indicates that either are possible within this event.

Major Comment #3: Line 299, 444 and elsewhere: Dendritic growth temperature zone employed here seems very narrow, often -20 to -10 °C are used instead -12 to -8°C, see e.g. https://acp.copernicus.org/articles/22/11795/2022/ and references therein.
Thanks for this comment! We used -12 to -18 C for the dendritic growth zone, but made a typo on line 444 misstating this. While we acknowledge the full dendritic growth temperature zone extends beyond the range used, we want to maintain more equal sized categories for the temperature analysis in Figure 7, as widely unequal category sizes would bias the histogram. References to the dendritic growth zone are now clarified to indicate that while the full range of temperatures can be considered the dendritic growth zone, we refer to the central temperatures of the dendritic growth zone to maintain roughly equally sized categories of 4 to 6 degrees.

Major Comment #4: In line 233 dendritic growth zone temperature regime is reported as -18 to -12°C – use consistent T-ranges throughout the manuscript.

Thanks for catching this! We used -12 to -18 C for the dendritic growth zone, but made a typo on line 444 misstating this. We will also now be clear when we are referring to the full range of the dendritic growth zone and when we are referring to the central temperatures of the dendritic growth zone within Figure 7.

Major Comment #5:  Section 3.1 (Case Verification) I think the study would benefit from adding two additional parameters "distance of multi-modal layer top from cloud top" (to see where the multimodalities occur with respect to cloud top) as well as cloud depth.

Thanks for this comment, we agree and this is being added to the revision. We looked into obtaining the cloud thickness and cloud top heights from the ARM value added products and are now using the cloud boundaries from the CLDTYPE Value Added Product (VAP). This product uses lidar and radar cloud boundaries obtained from the Active Remotely Sensed Cloud Location (ARSCL) and surface meteorological systems (MET) data. Incorporating this dataset has improved the case-based analysis as you described above, and also led to us incorporating this dataset into the replacement for the long-term analysis (see response below to comment #6 pertaining to replacing the ERA5 monthly analysis). We are adding the analysis of cloud tops and cloud thickness to Figure 3 and the associated discussion in Section 3.1 following the reprocessing of cases after adjustments to the detection criteria.

Major Comment #6: Line 190-194: Please explain why you choose to set monthly flag count into context with ERA-5 monthly thermodynamic, kinematic, and microphysical variables – Fig.4 shows that high flag counts are mostly related to single continuous events. Why not analyze the thermodynamic, kinematic, and microphysical variables for those events instead? Or instead contrast ERA-5 variables for flagged events vs. non-flagged cloudy periods? I struggle seeing the benefit of the correlation of the monthly multi-modal flag count with ERA-5 variables as it is presented unless it is set into context with existing literature, e.g. on seasonal mixed-phase cloud occurrence at the NSA site etc.

Thanks for this comment. We originally used the monthly values to assess if any potential patterns did exist on a broader, climatological scale as well as with consideration to the file size/computational burden when requesting and using hourly pressure level data. Ultimately, we agree that using these averaged quantities is not the best approach and we have decided to remove this analysis. To better investigate how atmospheric liquid and cloudiness correlates with events, we are replacing this analysis with the aforementioned analysis using observational datasets including precipitable water vapor from the MWR, liquid water path, cloud base/top heights, and cloud thickness.

Because of extended periods of missing KAZR data in 2022 and missing MWR data in 2023, we currently use only 2020 for this analysis. 2020 has only one date of missing KAZR data and 12 dates of missing MWR data.

The new analysis for this section contrasts the flagged events vs. non-flagged cloudy periods, as you suggested. Cloudy periods are determined using the micropulse lidar (MPL) and radar derived cloud fraction, from the nsaarmbecldradC1 product. We tested several thresholds of cloud fraction to determine if the hour is cloudy or not, and presently are using cloud fraction exceeding 75% to define a cloudy hour.

First, we analyzed the flagged hours and cloudy hours across 2020 (Fig. R. 5). This analysis found that when using a cloudiness threshold of cloud fraction exceeding 75%, a total of 6321 of 8784 hours in the year were cloudy. 16.6% of those cloudy hours (1057 hours) were also

flagged as exceeding 100 flags per hour (11.9% of all hours in the year). Interestingly, five hours were flagged that do not meet the cloudiness threshold. Note that this analysis examines all hours with flag count of 100 or greater, not just events that see persistent signals of 2 or more hours.

[Figure]

Fig R. 5: Matrix visualization of the hours of 2020 that are cloudy and that meet the 100+ flag criteria for an event hour.

Second, we examined observations of precipitable water vapor (PWV), liquid water path (LWP), cloud base, cloud top, cloud thickness, and hourly precipitation for hours with (1) greater than 100 criteria flags per hour and cloudy conditions and (2) under 100 criteria flags per hour and cloudy conditions. Because these observations do not conform to normal distributions, t-tests and Pearson correlations were not very conclusive, and we instead visualize the distributions of these two datasets with violin plots. This is shown in Fig. R. 6. As a general trend across this analysis, the flagged events had more moisture and deeper clouds. The cloud tops and thicknesses from this dataset were very broadly spread for both the flagged and unflagged cloudy periods. For cloud thickness both interquartile ranges were approximately 4000 m wide, which is largely related to the variability in cloud tops. This may be affected by high level cirrus clouds, rather than the tops of the specific cloud layers producing a multi-modal signal. We did retest this for varied definitions of cloudiness by total cloud cover with little impact on the results. As mentioned in your minor comment #6 further down this document, the case-based analysis does exclude single hour events. This new analysis incorporates those single hour events to include their moisture contents and cloud heights to better understand the flagged clouds compared to the unflagged clouds across a year.

[Figure]

Fig R. 6: Violin plots of observed quantities shown for hours in 2020 that contain 100 or more criteria flags per hour and hours in 2020 that do not meet this criteria, but do exceed 75% cloud cover. For events: the thick black line spans in the interquartile range and the thin grey line spans from the 10th to 90th percentiles. For no event: the thick green line spans in the interquartile range and the thin light green line spans from the 10th to 90th percentiles. (a) Precipitable water vapor as observed by MWR (b) liquid water path (c) cloud base (d) cloud top (e) cloud thickness (f) hourly liquid equivalent precipitation.

In panels (a) and (b) you can see the tendency for flagged periods to be moister than unflagged cloudy periods, though there is significant overlap between them. In panel (c), there is a clear difference in the cloud bases of flagged events and non-flagged cloudy periods. The interquartile range (IQR) of event hour cloud base is 84-1124 m and of non-event cloudy hours is

245-3189 m. If we instead consider average cloud base, the height for event hours is 942 m and non-event hours is 1886 m. This is potentially indicating that colder, higher clouds are not producing multi-modal signals and are not a significant portion of the detected events. Note this also lines up with the results shown in Figure 3, which for events of 2 or more hours showed a mean detected layer height near 1750 m and mean detected layer depth of approximately 1250 m.

In the last panel when examining hourly precipitation, there is a clear indication that flagged hours have greater hourly precipitation rate. This makes sense because the coexistence of multiple hydrometeor types (e.g. snow aggregates and cloud ice or graupel and liquid cloud droplets) would be expected to produce a multi-modal spectrum as the precipitation should typically has a faster downward vertical velocity (fall speed) than cloud liquid or cloud ice.

Thanks to both reviewers for comments on this analysis; we feel that the new iteration using hourly observations is a much stronger representation of the conditions associated with these signals.

**Minor comments:**

Minor Comment #1: Throughout the text, it would be helpful to always state which "flag" (Multi-modal flag/LDR flag) is meant to avoid confusion (e.g. Line 444f "the algorithm detected flags that aligned with the multi-modal layers".)
We agree and this clarification has been made.

Minor Comment #2: Line 10: please check the sentence, what does the word "verifying" refer to?
We understand this sentence could be confusing to readers, and have clarified it to read: "Based on three years of data, we show that the detection algorithm effectively identifies multi-modal spectra, with 90.8% of detected events verified." Note that scripts for the verification are being re-run and the exact percentage may change in the edited manuscript.

Minor Comment #3: Line 16 and elsewhere: Clearly state where you refer to spectral or integrated LDR
Thanks for catching this, we have added clarification for when we are referring spectral or moment LDR.

Minor Comment #4: Line 55: How does the manual verification work?
This is expanded on in lines 96-102, though we do see that it should be written with more detail. Additionally in response to comments on Part 1, a sub-section is being added to section 3 containing a one-month period that is examined using a Gaussian peak fitting algorithm as a secondary check for our statistics and to assess for missed events.

Minor Comment #5: Line 76: replace "being a secondary mode" with "containing a secondary mode"
This correction has been made.

Minor Comment #6: Line 80: Motivate why you choose two hours as minimum threshold for case identification? – Depending on cloud type and cloud lifetime, microphysical processes with pronounced multi-modal Doppler spectra occur on much shorter time scales.

The consideration that the processes generating secondary modes may exist on short time scales was considered in the original determination of the use of 100 flags. The threshold of 100/hour can be reached by an approximately 750 m deep multi-modal layer in just over 10 minutes for example. For a user who is applying the criteria and seeking short events, the algorithm will be able to identify those shorter periods. Identification of these shorter events could be done by partitioning the flag counts into periods shorter than an hour or examining the output in the original temporal resolution (two minutes). We can improve the way we discuss this in the manuscript and better indicate that the minimum length can be determined by a user for their specific study.

By creating a minimum of two sequential hours meeting the criteria, we were initially reducing the total amount of spectral data processing required to verify events/cases across three years. The added analysis described in response to Major Comment #6 does now consider all hours that are flagged, rather than only those that met the additional length criteria for verification and case analyses. This new analysis addresses much of this, albeit only for 2020 as there are periods of missing data across the various instrumentation 2022-2023.

Minor Comment #7: Line 88: Explain how this consolidation of cases is done. Manually?

The start and end hours of each detected case are initially determined by the first and final hour with 100 flags across that period, but after examining the CSV of these times and flag counts, it became apparent that some cases that were continuous were being split by near misses of the criteria of 100 flags per hour (e.g. 98 flags instead of 100), and so those were manually consolidated by extending the end hour. This ultimately results in a handful of longer cases and fewer instances of 2+ separate cases within a single date.

Minor Comment #8: Line 230-231: Please give references for the statement that "primary mode" refers to the faster falling one and "secondary mode" to the slower falling one. In KAZR terms, I think primary mode is the one with the highest reflectivity independent of fall velocity.

Thanks for this comment, we agree that primary mode should refer to the highest reflectivity mode. This sentence has been rephrased to indicate the primary refers to the higher reflectivity mode and subsequent primary/secondary references align with the correction.

Minor Comment #9: Line 234: Rephrase: "The distinction between liquid droplets and columnar ice/needles" …
Thanks for this edit, it makes the sentence much more clear. This has been rephrased.

Minor Comment #10: Line 282: add "columnar" before ice crystals
This correction has been made.

Minor Comment #11: Line 352: replace "profile" with "atmosphere"
This correction has been made.

Minor Comment #12: Line 431: Explain why would you limit the algorithm to vertically-pointing Ka-band radar observations and don't propose to also use it for other radar bands (X-/W-/G-band)?
This line was intended to restate that this analysis demonstrated utility in applying this to Ka-band radar observations, rather than directly limits application of this concept to radars of other wavelengths. Repetition of a similar method on another wavelength of radar may require tuning of the parameters used to compensate for differences that may arise in the magnitudes of spectrum width or signal-noise ratio adjustments. We have clarified this line.

Minor Comment #13: Line 434: add "columnar" before ice
This correction has been made.

Minor Comment #14: Line 448: As stated above, I strongly suggest including additional remote-sensing observations existing at ARM KAZR sites
See comments above to Major Comment #1 and Major Comment #6 where additional observations have been added to supplement what we can glimpse from KAZR.

**Comments on Figures:**
Fig. 1: Please add time-height panels of KAZR reflectivity, mean Doppler velocity, and spectrum width to contextualize the two case studies for which the flag-criteria depicted are matched. In this way the readers can also see if vertical spectrograms are applicable or if Doppler spectrum evolution seems rather plausible along slanted fall streaks.
Thanks for this comment! This is being added to Figure 1 as well as the discussion of these examples. We did note that in the original example of a streak, the spectrum width has artifacts from a potential problem with the radar. This may be related to the artificial broadening noted in 2021 that was likely caused by a malfunctioning phase lock oscillator (see lines 67-69 in the preprint), though this oscillating broadening is not detected by the detection algorithm.

[Figure]

Fig R. 7: (a) For 19 October 2020: (i) Time-height depiction of radar gates that met the SW criterion (pink shading), the MDV criterion (blue shading), and where both criteria are met (black shading), indicating regions containing flags identified through the multi-modal detection algorithm, (ii) radar reflectivity, (iii) spectrum width, and (iv) vertical velocity. (b) As in (a) but for 31 October 2023.

Fig 1: Is Oct 31 2023 (title) or 2024 (caption) shown?
Apologies for the typo, the dates should read 19 October 2020 and 31 October 2023. This has been corrected.

Section 5: Add time-height plots of KAZR reflectivity, MDV, spectrum width and LDR for all three presented case studies to contextualize the presented height spectrograms.
These figures have been added to the revised copy of the paper and are attached below.

[Figure]

Fig R. 8: Radar moment variables for the 15 April 2020 case: (a) reflectivity (b) linear depolarization ratio (c) spectrum width (d) vertical velocity.

[Figure]

Fig R. 9: Radar moment variables for the 3 September 2020 case: (a) reflectivity (b) linear depolarization ratio (c) spectrum width (d) vertical velocity.

[Figure]

Fig R. 10: Radar moment variables for the 4 January 2022 case: (a) reflectivity (b) linear depolarization ratio (c) spectrum width (d) vertical velocity.

Fig 10: What are the horizonal lines in panel d, e, f? (also in Fig. 12)
These horizontal lines in those panels are artifacts of the data quality; the reflectivity and LDR both have values of "nan" at those heights pictured as white horizontal stripes. This can be resolved by altering our spectra figures to a short temporal average rather than instantaneous spectra. There are trade-offs to both choices, but we are altering the figures to use averages over twenty seconds, which reduces the noise and erroneous lines. An example of this is attached in Fig. R. 11. The final result is still not perfect (note the line near 2 km), but it is clearer to readers.

[Figure]

Fig. R. 11: Comparison of (a) instantaneous spectra and (b) 20 s average spectra from 03 September 2020 at 2355 UTC. The instantaneous spectra appeared in the original submission as Fig. 10 (f).

Fig 10: extend your x-axis to lower velocities to capture the entire observed Doppler spectra
This correction has been made, shown in Fig. R. 11 above.

---

## Author Comment (AC2)

Reviewer #2

**General Comments**
General comment #1: The standard abbreviation of the 5th generation ECMWF reanalysis is
"ERA5". Thus, all instances of "ERA-5" throughout the manuscript should be revised.
Thank you for pointing this out, this correction has been made in the manuscript.

General comment #2: Relatedly, the only major concern I have about the construction of the effort
relates to the reliance on monthly mean ERA5 fields for the broad correlation analysis used. The
precipitation events comprising the multimodal cases occur sporadically throughout the year and
are seldom evenly distributed throughout the month. Reducing the information content of the
reanalysis to a monthly mean would seem to be counterproductive to identifying prevailing
relationships with environmental characteristics and the cases evaluated. Namely, I would imagine
that in some months a multimodal case could occur within an environmental extreme compared to
the monthly mean and thus the correlations assessed would be mostly meaningless. I recommend
utilizing the higher-rate (i.e., hourly) and higher-resolution ERA5 output sampled representatively
from each event as the basis for this work.

       Thanks for this comment. We originally used the monthly values to assess if any
potential patterns did exist on a broader, climatological scale as well as with consideration to the
file size/computational burden when requesting and using hourly pressure level data. Ultimately,
we agree that using these averaged quantities is not the best approach and we have decided to
remove this analysis. To better investigate how atmospheric liquid and cloudiness correlates with
events, we are replacing this analysis with the aforementioned analysis using observational
datasets including precipitable water vapor from the MWR, liquid water path, cloud base/top
heights, and cloud thickness.

       Because of extended periods of missing KAZR data in 2022 and missing MWR data in
2023, we currently use only 2020 for this analysis. 2020 has only one date of missing KAZR
data and 12 dates of missing MWR data.

       The new analysis for this section contrasts the flagged events vs. non-flagged cloudy
periods, as Reviewer 1 suggested. Cloudy periods are determined using the micropulse lidar
(MPL) and radar derived cloud fraction, from the nsaarmbecldradC1 product. We tested several
thresholds of cloud fraction to determine if the hour is cloudy or not, and presently are using
cloud fraction exceeding 75% to define a cloudy hour.

       First, we analyzed the flagged hours and cloudy hours across 2020 (Fig. R. 1). This
analysis found that when using a cloudiness threshold of cloud fraction exceeding 75%, a total of
6321 of 8784 hours in the year were cloudy. 16.6% of those cloudy hours (1057 hours) were also
flagged as exceeding 100 flags per hour (11.9% of all hours in the year). Interestingly, five hours
were flagged that do not meet the cloudiness threshold. Note that this analysis examines all hours
with flag count of 100 or greater, not just events that see persistent signals of 2 or more hours.

[Figure]

Fig R. 1: Matrix visualization of the hours of 2020 that are cloudy and that meet the 100+ flag criteria for an event hour.

Second, we examined observations of precipitable water vapor (PWV), liquid water path (LWP), cloud base, cloud top, cloud thickness, and hourly precipitation for hours with (1) greater than 100 criteria flags per hour and cloudy conditions and (2) under 100 criteria flags per hour and cloudy conditions. Because these observations do not conform to normal distributions, t-tests and Pearson correlations were not very conclusive, and we instead visualize the distributions of these two datasets with violin plots. This is shown in Fig. R. 2. As a general trend across this analysis, the flagged events had more moisture and deeper clouds. The cloud tops and thicknesses from this dataset were very broadly spread for both the flagged and unflagged cloudy periods. For cloud thickness both interquartile ranges were approximately 4000 m wide, which is largely related to the variability in cloud tops. This may be affected by high level cirrus clouds, rather than the tops of the specific cloud layers producing a multi-modal signal. We did retest this for varied definitions of cloudiness by total cloud cover with little impact on the results.

[Figure]

Fig R. 2: Violin plots of observed quantities shown for hours in 2020 that contain 100 or more criteria flags per hour and hours in 2020 that do not meet this criteria, but do exceed 75% cloud cover. For events: the thick black line spans in the interquartile range and the thin grey line spans from the 10th to 90th percentiles. For no event: the thick green line spans in the interquartile range and the thin light green line spans from the 10th to 90th percentiles. (a) Precipitable water vapor as observed by MWR (b) liquid water path (c) cloud base (d) cloud top (e) cloud thickness (f) hourly liquid equivalent precipitation.

     In panels (a) and (b) you can see the tendency for flagged periods to be moister than unflagged cloudy periods, though there is significant overlap between them. In panel (c), there is a clear difference in the cloud bases of flagged events and non-flagged cloudy periods. The interquartile range (IQR) of event hour cloud base is 84-1124 m and of non-event cloudy hours is

245-3189 m. If we instead consider average cloud base, the height for event hours is 942 m and non-event hours is 1886 m. This is potentially indicating that colder, higher clouds are not producing multi-modal signals and are not a significant portion of the detected events. Note this also lines up with the results shown in Figure 3, which for events of 2 or more hours showed a mean detected layer height near 1750 m and mean detected layer depth of approximately 1250 m.

In the last panel when examining hourly precipitation, there is a clear indication that flagged hours have greater hourly precipitation rate. This makes sense because the coexistence of multiple hydrometeor types (e.g. snow aggregates and cloud ice or graupel and liquid cloud droplets) would be expected to produce a multi-modal spectrum as the precipitation should typically has a faster downward vertical velocity (fall speed) than cloud liquid or cloud ice.

Thanks to both reviewers for comments on this analysis; we feel that the new iteration using hourly observations is a much stronger representation of the conditions associated with these signals.

With the additional datasets used to replace the original analysis as well as to supplement the cases examined in more detail, citations for additional ARM observational datasets and value added products (VAP) are attached:

Cadeddu, Maria, et al. "Microwave Radiometer Profiler (MWRP), 2004-02-19 to 2023-09-28, North Slope Alaska (NSA), Central Facility, Barrow AK (C1)." Atmospheric Radiation Measurement (ARM) User Facility, doi:10.5439/1025254. Accessed 25 May 2025.

Cromwell, Erol, et al. "Precipitation Imaging Package (PRECIPIPMP), 2018-10-23 to 2025-04-21, North Slope Alaska (NSA), Central Facility, Barrow AK (C1)." Atmospheric Radiation Measurement (ARM) User Facility, doi:10.5439/1489525. Accessed 25 May 2025.

Chen, Xiao, and Shaocheng Xie. "ARM Best Estimate Data Products (ARMBECLDRAD), 1998-01-01 to 2023-12-31, North Slope Alaska (NSA), Central Facility, Barrow AK (C1)." Atmospheric Radiation Measurement (ARM) User Facility, doi:10.5439/1333228.

Zhang, Damao, et al. "Cloud Type Classification (CLDTYPE), 1998-03-25 to 2023-08-30, North Slope Alaska (NSA), Central Facility, Barrow AK (C1)." Atmospheric Radiation Measurement (ARM) User Facility, doi:10.5439/1349884.

General comment #3: Though it could be argued to be beyond the scope of the study, an example with collocated aircraft data would be a tremendously meaningful addition to the manuscript, primarily as a more robust validation exercise for the algorithm. I recognize that such data may not exist for the site used, but it would be helpful for that to be acknowledged as justification for not executing such work, should that be the case.

Thanks for this comment, if aircraft data for these years did exist at the NSA site it would be immensely useful. Within our conclusions, we will now note that the repetition of this with aircraft data from a field campaign directly over a Ka-band radar would be advantageous for better understanding the observations and associated signals and attributing processes.

**Specific Comments**

Line 13: "access" should be "assess"
Thank you for pointing this out, this correction has been made.

Lines 36-39: this is a complicated and confusing sentence. Please clarify and split into 2 sentences.
Apologies for this, the sentence is now split into two and clarified.

Line 150: simplify "the vast majority of" to "most"
Thank you for pointing this out, this correction has been made.

Lines 233-234: this was already said. Add citations to previous mention or delete.
Thank you for pointing this out, this has been consolidated with the previous mention.

Figure 13: this image is very blurry. Please improve resolution.
Apologies for this, we regenerated the figure and this is now corrected.

Line 439: Unnecessary comma after "cases"
Thank you for pointing this out, this correction has been made in the revision.

---

## Author Response (AR2)

**Author Response: Changes to the Manuscript**

Responses to Reviewer #1:

1. In the new figures 12, 16, 20, note more clearly which variable is derived from MWR and from PIP, either in the caption or in the subfigure titles.

Thank you for this comment, the captions have been updated. The new captions are as follows:

Figure 12: Additional observations of precipitable water and liquid water content from the MWR and hydrometeor size distribution from the PIP on 15 April 2020 (a) precipitable water content (b) liquid water content (c) distribution of hydrometeor density vs. diameter.

Figure 16: Additional observations of precipitable water and liquid water content from the MWR and hydrometeor size distribution from the PIP on 3 September 2020 (a) precipitable water content (b) liquid water content (c) distribution of hydrometeor density vs. diameter. Note the different y-axis scale in panel (c) compared to Fig. 12.

Figure 20: Additional observations of precipitable water and liquid water content from the MWR and hydrometeor size distribution from the PIP on 4 January 2022 (a) precipitable water content (b) liquid water content (c) distribution of hydrometeor density vs. diameter.

2. New Fig 5: title of subfigure f) should be precipitation rate

Thank you for this correction, the plot title has been adjusted in the revision.

3. New Fig R3,4,5: Please check the linear depolarization ratio subfigures. Commonly, LDR is filtered with signal-to-noise-ratios (SNR) of 5-10 dB. Was SNR-filtering applied here?

Thank you for this comment; for the purpose of identifying events with significant LDR was subject to an SNR constraint, and so a mask of SNR>5 dB is now applied to the plotted variables in those figures. New figures now look as in below:

Responses to Reviewer #2:

The authors have done a nice job responding to previous reviewer comments. The revised analyses and discussion have certainly improved the paper. I have identified a handful of required technical edits, outlined below.

Line 30: "with" should be "when"

Line 114: "available routinely" would be better as "routinely available"

Line 143: November should also be mentioned – it is higher than May, which is mentioned.

Line 165: "ranging from" would be better as "spanning"

Line 209: delete "very"

Line 224: "potential" should be "potentially"

Line 522: "expenses of" should be "expense for"

Thank you for these corrections, all have been made.

Several figures remain blurry, especially Figure 19. I am not sure the reason, but it should be resolved prior to print.

Thank you for pointing this out, to address this the paper has been reformatted into LaTeX to avoid Microsoft Word scaling the original image files. Additionally they have be reproduced with DPI increased to 350.